

# A solid-state IR laser for two-step desorption/ionization processes in single-particle mass spectrometry

Marco Schmidt[1,2,3], Haseeb Hakkim[1,2,3], Lukas Anders[1,2,3], Aleksandrs Kalamašņikovs[1,2,3], Thomas Kröger-Badge[1,2,3], Robert Irsig[3,4], Norbert Graf[5], Reinhard Kelnberger†[5], Johannes Passig[1,2,3] and Ralf Zimmermann[1,2,3]

[1]Joint Mass Spectrometry Centre, Analytical Chemistry, University Rostock, 18059 Rostock, Germany
[2]Joint Mass Spectrometry Centre, Helmholtz Zentrum München, 85764 Neuherberg, Germany
[3]Department Life, Light & Matter, University of Rostock, 18051 Rostock, Germany
[4]Photonion GmbH, 19061 Schwerin, Germany
[5]InnoLas Laser GmbH, 82152 Krailling, Germany
† deceased

*Correspondence to*: Johannes Passig (Johannes.Passig@uni-rostock.de)

**Abstract.** Recent advancements in single-particle mass spectrometry (SPMS) have enabled the detection of aromatic hydrocarbons at the individual particle level in conjunction with inorganic/refractory particle components. However, the laser desorption (LD) of organic material from particles prior to their ionization in a two-step process necessitates pulsed infrared lasers with adequate pulse energy that can be irregularly triggered on detected particles. Pulsed $CO_2$ lasers with a 10.6 μm wavelength have been traditionally utilized, yet these lasers are bulky, costly, and require regular maintenance, including gas exchange or a continuous laser gas supply. In this study, we present the application of a prototype solid-state laser based on an erbium-doped yttrium aluminum garnet (Er:YAG) crystal, emitting long pulses of 200 μs at 3 μm wavelength as a compact, cost-effective, and user-friendly alternative for LD. We directly compared the new laser with a commonly used $CO_2$ laser and found similar performance in LD for both laboratory particles and ambient air experiments. With the exception of slightly increased fragmentation observed with the $CO_2$ laser due to its beam profile, no qualitative differences were noted in the resulting mass spectra. Additionally, we compared the novel two-step ionization scheme for the combined detection of aromatic molecules and inorganics with conventional single-step laser desorption/ionization (LDI) for the detection of polycyclic aromatic hydrocarbons (PAH) in laboratory and field experiments. The combined methods demonstrated superior performance in the detection of PAHs, for both the $CO_2$ and the new Er:YAG laser. In addition to its higher sensitivity and lower fragmentation for PAHs when compared to single-step LDI, it is less dependent on the particle matrix, sharing the benefits of traditional two-step methods but extending its capability to combine PAH measurements with the LDI-based detection of inorganic particle compounds.



**1 Introduction**

Single-particle mass spectrometry (SPMS) has significantly expanded our understanding of aerosols and atmospheric processes due to two inherent characteristics of the technique. (I) By desorbing and ionizing individual particles using laser pulses, SPMS reveals the mixing state of the particle ensemble, i.e., the distribution of chemicals within individual particles. This allows the detection of subpopulations of particles, their sources, and the investigation of atmospheric processes in complex aerosols. (II) Unlike prevalent bulk methods based on thermal desorption, such as the Aerodyne Aerosol Mass Spectrometer (AMS) or proton transfer reaction-based methods for aerosols (Reinecke et al., 2024a, 2024b), SPMS also captures refractory particle compounds like metals and mineral dust components (Pratt and Prather, 2012; Passig and Zimmermann, 2021; Laskin et al., 2018; Marsden et al., 2019; Zawadowicz et al., 2017; Passig et al., 2020).

In most SPMS instruments, particles are hit by intense, focused UV laser pulses to create a sufficient ion signal for less abundant particle components via laser desorption/ionization (LDI). With few exceptions, organic molecules are fragmented in this process, making their speciation difficult. In contrast to the use of single UV laser pulses for LDI, so-called two-step approaches first use an IR laser pulse to vaporize the organic matter from the particle, and a second UV laser pulse to hit the resulting plume and ionize the molecules in the gas phase. The separation of laser desorption (LD) and ionization allows the independent optimization of both processes. This has several advantages: while LD alone is still a complex process resulting in the vaporization of molecules at high temperatures, the laser intensities and photon energies can be much lower compared to LDI, reducing fragmentation (Schmidt et al., 2023). Because ionization occurs in the gas phase, matrix effects are reduced which is beneficial for quantifying approaches (Woods et al., 2001). In addition, gas-phase ionization allows for very soft ionization techniques such as single-photon ionization (SPI) (Nash et al., 2005; Hanna et al., 2009) or resonance-enhanced multiphoton ionization (REMPI) (Gehm et al., 2018). The latter is particularly useful for SPMS because the involved resonances make it very sensitive – a key characteristic when dealing with such small sample volumes as plumes surrounding tiny particles – and it selectively ionizes aromatic molecules with very high efficiency. This is of great benefit for aerosol studies because it detects carcinogenic polycyclic aromatic hydrocarbons (PAHs), ubiquitous air pollutants from combustion processes, and major contributors to the health effects of aerosols (Agudelo-Castañeda et al., 2017; Holme et al., 2019). The two-step LD-REMPI method, introduced in SPMS by Morrical et al. (1998), has been used by a few groups to study PAHs at the single-particle level, e.g., to identify sources of PAHs in ambient air pollution (Bente et al., 2009) or to analyze particles for studying aging effects (Li et al., 2019). One significant disadvantage of these two-step approaches is the missing particle composition from LDI, i.e., the metals, salts, etc.

This constraint has been successfully addressed, first by implementing a series of three laser pulses for LD, REMPI, and LDI in a rather complex setup that enables the measurement of only positive ions (Passig et al., 2017). Meanwhile, a novel technique has been devised to leverage the spatial segregation of the gas plume from the particle residue, enabling the integration of REMPI and LDI into a single laser pulse with a customized radial profile and yielding bipolar LDI spectra in addition to the





PAH signatures (Schade et al., 2019; Zimmermann et al., 2019). The latter method has been successfully applied in ambient air studies and laboratory experiments, showing its potential for source apportionment, new monitoring concepts, and investigations of atmospheric processes (Passig et al., 2022; Anders et al., 2023; Anders et al., 2024). There are also hybrid

approaches where LD is applied prior to LDI at 193 nm to reduce fragmentation of organics while providing LDI mass spectra comparable to many other instruments (Zelenyuk et al., 2015).

To date, all two-step methods in SPMS are based on transversely excited atmospheric pressure (TEA) $CO_2$ lasers, because these systems provide relatively strong and short mid-IR pulses (multi-mJ, ≈ 50–500 ns duration, 10.6 µm wavelength). These lasers are relatively large and expensive, and they have one major disadvantage: they require regular gas changes or even

operation with a constant gas flow. The required gas supply limits the use of $CO_2$-TEA lasers in field studies, especially in situations where aerosol measurements are urgently needed: in arctic regions, on the open sea, in remote areas with limited infrastructure, and for remote controlled long-term operation. For the ionization step in two-step approaches, frequency-quadrupled Nd:YAG lasers at 266 nm offer a low-cost, robust, and maintenance-free alternative to excimer gas lasers. However, a similar solution is still lacking for the LD step.

Here we report on experiments to replace the $CO_2$ lasers with a prototype Er:YAG solid-state laser at 3 µm wavelength. We evaluate its performance in direct comparison experiments and discuss its potential and limitations in future applications. Besides, we performed comparative experiments and demonstrated superior detection of PAHs using two-step approaches compared to conventional LDI, regardless of the type of laser used for LD.

## 2 Experimental Section

### 2.1 Model Particles and Sampling


Laboratory experiments were performed for three different types of PAH-containing particles. (I) Diesel exhaust particles were collected from an old van (VW Transporter type 3, 1.7 D, no exhaust aftertreatment), directly scratched from the inner surface of the exhaust pipe. These particles have a uniform chemical distribution of PAHs from regular exposure to the hot exhaust gas (Passig et al., 2017; Schade et al., 2019). (II) Wood ash particles were collected from an ash sink in the chimney of a 20

kW wood combustion furnace (Bente et al., 2008). Using a small-scale powder disperser (model 3433 SSPD, TSI Inc., St. Paul, MN), diesel soot and wood ash particles were redispersed into a synthetic air stream of 2.5 L min$^{-1}$, from which 100 mL min$^{-1}$ was introduced into the SPMS system. (III) Tar ball particles were used as a proxy for organic aerosols (Li et al., 2019). The term tar ball refers to near-spherical, homogeneous particles resulting from the combustion of biomass and biofuels and consisting of amorphous carbonaceous species (Pósfai et al., 2004; Hand et al., 2005; Alexander et al., 2008). To produce

the tar ball particles, beechwood tar from a hunting supply store was dissolved in methanol, sprayed with an aerosol generator





(ATM 221, Topas GmbH, Germany), and dried using a custom diffusion dryer with a silica gel-filled cartridge. Size distributions of the particles are shown in the supplementary information, Fig. S1.

The ambient air experiments were performed at the ILMARI laboratory of the University of Eastern Finland in the timeframe from March 1, 2024 to March 3, 2024. To record a sufficient number of single-particle spectra from the relatively clean air

($PM_{10} < 4.0 \ \mu g \ m^{-3}$ in average), ambient air particles were sampled using an aerosol concentrator (AC-250 v1.0, ParteQ GmbH, Germany). The instrument is based on an advanced design of the discontinued Model 4240 (MSP Corp., now part of TSI Inc., U.S.A.) and concentrates particles in the size range of approximately 0.5–10 µm from a 250 L $min^{-1}$ inlet flow into a 1 L $min^{-1}$ sample stream (Romay et al., 2002). After passing through a dryer (model MD-700-12S-1, Perma Pure LLC, USA), the aerosols underwent additional concentration to 0.1 L/min through a further virtual impactor at the SPMS aerodynamic lens

(Zhuo, Z., Su, B., Xie, Q., Li, L., Huang, Z., Zhou, Z., Mai, 2021). The instrument's overall detection efficiencies for the particles used in this study were not determined (Shen et al., 2019).

### 2.2 SPMS Instrumentation

The basic principle of SPMS systems relies on the optical detection and sizing of aerodynamically accelerated particles before LDI and ion detection in a bipolar time-of-flight mass spectrometry setup (Passig and Zimmermann, 2021; Pratt and Prather,

105 2012).

The instrument utilized here has been previously described (Schade et al., 2019) and roughly corresponds to the setup commercialized as PhotonLIZA (Photonion GmbH, Germany). A KrF excimer laser (248 nm, PhotonEx, Photonion GmbH, Germany) is used for ionization after laser desorption. The ion signals, from both cations and anions, were recorded by a 14-bit digitizer card (ADQ14, Teledyne SP Devices AB, Sweden) and were analyzed using custom Matlab software, including

preprocessing and peak area integration. Classification of the PAH mass spectra from ambient air particles was performed using the adaptive resonance theory neural network ART-2a (Song et al., 1999) from the open-source toolkit FATES (Sultana et al., 2017) using a vigilance factor of 0.75, a learning rate of 0.05, and 20 iterations.

### 2.3 Optical setup and triggering

In the two-step mode, the particles entering the ion source are exposed to an IR laser pulse for LD, and the gaseous plume is

then irradiated with an unfocused UV pulse of medium intensity for soft REMPI of aromatic species. This layout is illustrated in Fig. 1a and was used to study the LD with the Er:YAG laser in comparison to the $CO_2$ laser for the laboratory particles and the first of two ambient air experiments. To allow a direct comparison of the prototype Er:YAG laser (Er:YAG 1000 FQ, InnoLas Laser GmbH, Germany) with the standard $CO_2$ laser (Model EX5, GAM Laser Inc., USA), the setup was modified as follows: the two IR lasers were alternately fired from opposite sides of the ion source at the incoming particles; see Fig. 1a and

the time cycle included therein. A noise signal characteristic for each laser was imprinted in the mass spectra to ensure the



assignment of each laser to the resulting single-particle spectrum. Notable, that the instrument in this configuration could not record individual particle size information but only average size distributions.

The $CO_2$ laser and the Er:YAG laser vary not only in their wavelengths but also by several orders of magnitude in their pulse lengths; see Table 1 for detailed optical parameters. This is due to the lack of Q-switching. Within the 200 µs pulse duration,

the particles travel a distance of approximately 1-2 cm. Therefore, the laser is not focused but sent, with its full spot size of 5 mm, into the region above the ion source. The reduced spatial and temporal accuracy requirements facilitate alignment in comparison to the $CO_2$ laser. The lower power density of the Er:YAG pulse is compensated for by the longer exposure time for each individual particle and by the 10 times higher pulse energy. It should be noted, that the pulse-to-pulse energy variation of the Er:YAG laser was large (up to ± 50 % calculated on the average pulse energy, see Fig. S2). This is a result of unstable

thermal lensing in the resonator, as this prototype laser was not optimized for random triggering from the incoming particle events. To reduce this, a regular signal of 5 Hz was mixed into the trigger input. The total reduction in hit rate resulting from the combined loss in duty cycle and pulse-to-pulse variations is estimated to be ≈ 30–50 %, compared to a hypothetical stable Er:YAG laser to be developed for irregular triggering in SPMS. However, this is only a rough estimation because the pulse energy required to generate a sufficient plume is highly dependent on the particle properties and composition.

**2.4 Implementation in the combined ionization scheme for PAHs and inorganics**

At the end of this paper, we present an additional experiment on ambient aerosols to demonstrate the suitability of the Er:YAG laser for the newly developed ionization in SPMS that combines REMPI and LDI in a single laser pulse (Schade et al., 2019), and we compare this method directly with a conventional LDI-based setup. In the combined setup (Fig. 1b), the unfocused beam inducing REMPI with moderate laser intensity is reflected by 180°, is focused, and sent back into the ion source. There,

it hits the refractory particle residue, inducing LDI at virtually the same time as the PAHs are ionized in the gas plume via REMPI. The comparison with conventional LDI was realized as follows (Fig. 1, bottom): One particle event triggered the $CO_2$ laser for LD, followed by the excimer laser for combined ionization. The second particle triggered the same scheme with the Er:YAG laser instead of the $CO_2$ laser for comparison. The third particle triggered none of the LD lasers but only the excimer laser. As no LD occurred and no gas plume was produced, only the back-reflected beam induced LDI, similar to conventional

LDI ionization, with an excimer laser focused by a lens. All triggering schemes were realized using a Complex Programmable Logic Device (CPLD) microcontroller and custom software.



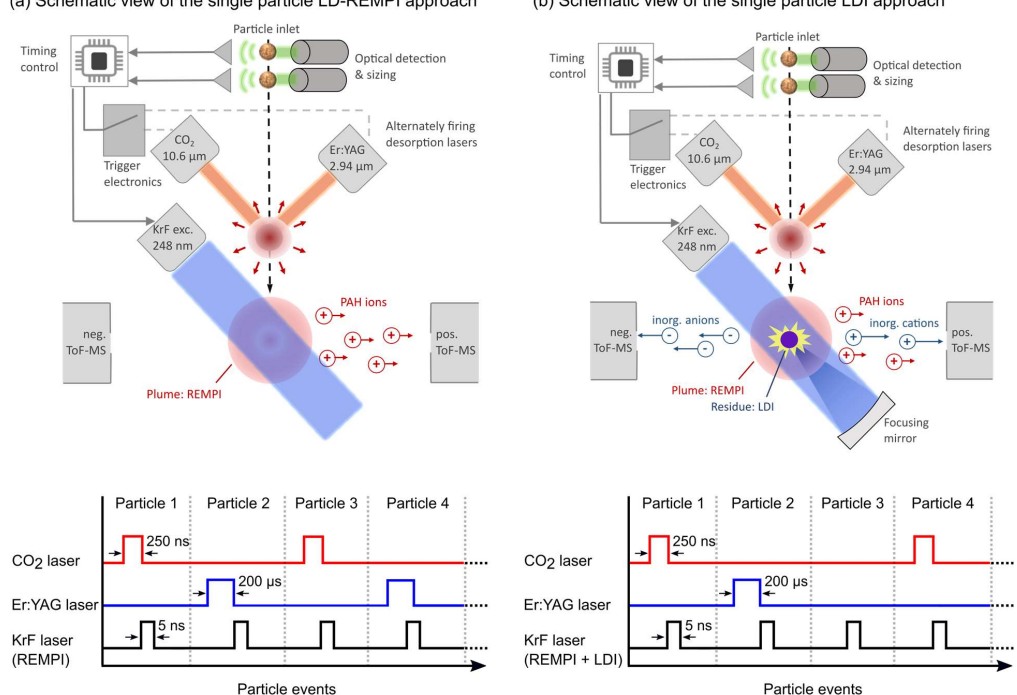

**Figure 1 (a)** Schematic representation of the single-particle, two-step ionization setup in alternating mode for direct comparison of the two desorption lasers. The particles are optically detected, initiating the SPMS time cycle. The two desorption lasers, i.e., the "standard" $CO_2$ laser and the experimental Er:YAG laser, are alternately fired at the respective particles, generating a small gas plume for each particle. The subsequent unfocused UV laser pulse ionizes the PAHs in the expanding plume via REMPI. Scheme **(b)** shows the combined method to ionize PAHs by REMPI in the unfocused beam and inorganics by LDI in the back-reflected and focused beam, both from the same laser. In addition to the comparison of the $CO_2$ laser and the Er:YAG laser for LD in this method, every third particle was not exposed to LD. In this case, only the intense back-reflected UV pulse interacted with the particle, inducing LDI without prior LD, allowing a direct comparison of the combined method with conventional LDI – also for ambient aerosols and on the same instrument.



**Table 1** Optical parameters and lasers systems used in the experiments.

|  | Laser desorption |  | REMPI | LDI |
|---|---|---|---|---|
| **Laser type, medium** | Gas, $CO_2$ | Solid-state, Er:YAG | Gas-Excimer, KrF | Gas-Excimer, KrF |
| **Wavelength (nm)** | 10600 | 2940 | 248 | 248 |
| **Beam diameter (mm)** | 1 x 1 (Gaussian) | 5 x 5 (Flat top) | 5 x 10 (Gaussian x flat top) | 0.2 x 0.4 (Gaussian x flat top) |
| **Pulse energy (mJ)** | 15 | 160 | 4 | 4 |
| **Pulse duration** | 250 ns | 200 µs | 5 ns | 5 ns |
| **Peak irradiance ($W\ cm^{-2}$)** | $1.5 \times 10^7$ | $3.2 \times 10^3$ | $1.6 \times 10^6$ | $1.0 \times 10^9$ |

## 3 Results and discussion

### 3.1. Two-step LD-REMPI for laboratory particles

First, we investigated the suitability of the Er:YAG laser for the LD-REMPI scheme (Fig. 1a) in laboratory experiments for

three different PAH-containing particle types, named diesel soot particles, wood ash particles, and tar ball particles. The most

probable PAHs reflected in the mass spectra are listed in Table 2. Note that isobar substances, *e.g.,* phenanthrene *vs.* anthracene,

*cannot be distinguished.* Figure 2a shows the sum PAH mass spectra of each 500 diesel soot particles, above with LD using

the $CO_2$ laser and below with the Er:YAG laser. The mass spectrometric patterns for both LD lasers are very similar, showing

detailed PAH signatures with dominant parent PAHs of low molecular weight from the combustion process (Frenklach, 2002)

and some alkylated species, partly consisting of residues from unburned fuel (Spencer et al., 2006; Anders et al., 2023). Note,

that the fragments are virtually unrecognizable, underscoring the soft nature of the LD-REMPI approach. The histograms on

the right show the distribution of PAH signal intensities among the particles for the two lasers. The $CO_2$ laser produces very

intense PAH mass spectra more frequently, while with the Er:YAG laser more particles with low to moderate signals are

observed. This can be attributed to the laser spot geometry: The Er:YAG laser has a large spot and hits the particle with high

probability but relatively low intensity, whereas the focused Gaussian profile of the $CO_2$ laser can result in very high signal

intensities when a particle is fully hit. However, the difference is rather moderate, also because of a comparable hit rate for

both lasers: 38 % (49 %) of the optically detected particles produced a clear PAH spectrum in this LD-REMPI approach with

the $CO_2$ laser (Er:YAG laser). It is also remarkable, that the signal intensities for the summed mass spectra of the same number

of particles are comparable for both lasers, despite the very different laser intensities, durations, and wavelengths. The high

degree of similarity can be explained by the underlying physical mechanisms of LD, where the energy transferred to the particle

eventually results in thermal desorption for laser pulses longer than a few picoseconds (Schmidt et al., 2023).



**Table 2** Polycyclic aromatic hydrocarbons (PAHs) indicated by the measured mass spectra.

| PAHs | m/z | | | | | |
|---|---|---|---|---|---|---|
| | Number of C in aliphatic side chains | | | | | |
| | 0 | 1 | 2 | 3 | 4 | 5 |
| Phenanthrene, anthracene | 178 | 192 | 206 | 220 | 234 | 248 |
| Pyrene, fluoranthene | 202 | 216 | | | | |
| Chrysene(s), benzoanthracene(s), benzophenanthrene(s) | 228 | 242 | 256 | 270 | 284 | |
| Benzopyrene(s), benzofluranthene(s), perylene | 252 | | | | | |
| Benzo[g,h,i]perylene, indeno[1,2,3-c, d]pyrene | 276 | | | | | |
| Dibenzophenanthrenes(s), dibenzoanthracene(s) | 278 | | | | | |
| e.g. Coronen | 300 | | | | | |
| e.g. Dibenzopyrene(s) | 302 | | | | | |
| Important fragments | 165, 189, 205, 219 | | | | | |

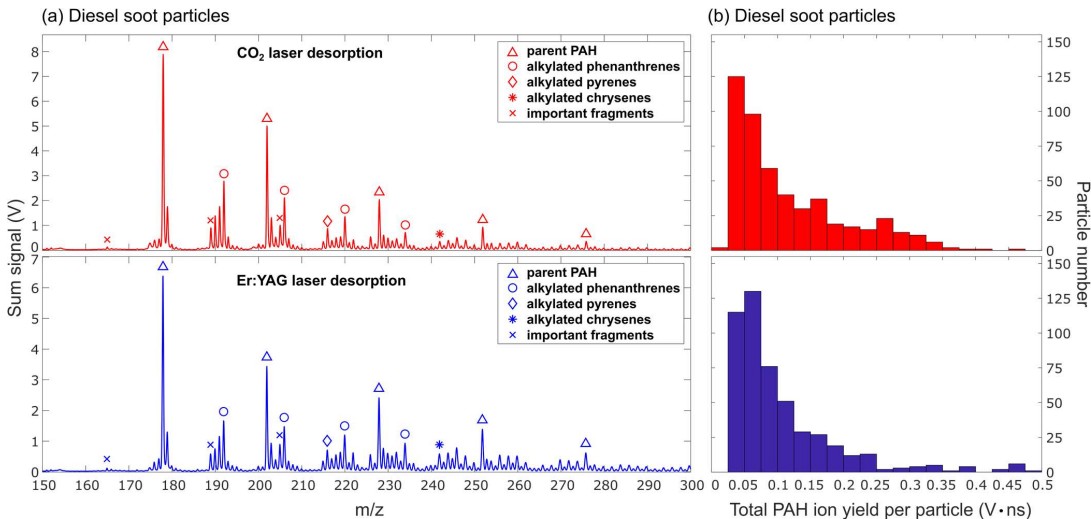


**Figure 2 (a)** The summed PAH mass spectra of each 500 diesel soot particles exhibit no significant differences when generated by LD-REMPI using a $CO_2$ laser (top) or the prototype Er:YAG laser (bottom) for the LD step. Note the very low fragmentation of the LD-REMPI method. **(b)** Histogram of the total PAH ion yield per particle. The $CO_2$ laser produces some more single-particle spectra with very strong PAH signals due to its high peak intensity in its Gaussian beam profile.





As a second representative particle type, wood ash particles were investigated. Here, the hit rate for PAHs was much lower, with only 2 % (4 %) of the optically detected particles showing a PAH spectrum when the $CO_2$ laser (Er:YAG laser) was used. This is due to the nature of the sample, which contains many burnt ash particles and fewer OC/soot particles containing PAHs (Dall'Osto et al., 2016; Healy et al., 2015). The PAH mass spectra are very different from the PAH signatures of diesel soot particles, highlighting the potential of single-particle PAH detection for source apportionment. Given the completely different

fuel, this is not surprising. In addition to parent PAHs as combustion products from the hydrogen-abstraction carbon-addition (HACA) mechanism (Frenklach, 2002), decomposition products from the biomass can also be detected, e.g., a strong signal for retene (m/z = 234). This is a thermal degradation product of resin acids and has often been used as a marker for softwood combustion (Ramdahl, 1983; Shen et al., 2012a) (The stove was mainly fired with spruce logs.). There are also some signals from higher-mass molecules, possibly from oxidized PAHs and other combustion products with aromatic rings, which can be

ionized in the LD-REMPI scheme. The comparison between the two LD lasers shows no clear differences. The histogram of single-particle PAH signal strengths on the right indicates again a slightly higher number of particles with very strong PAH signatures, probably due to the beam profile differences discussed above.

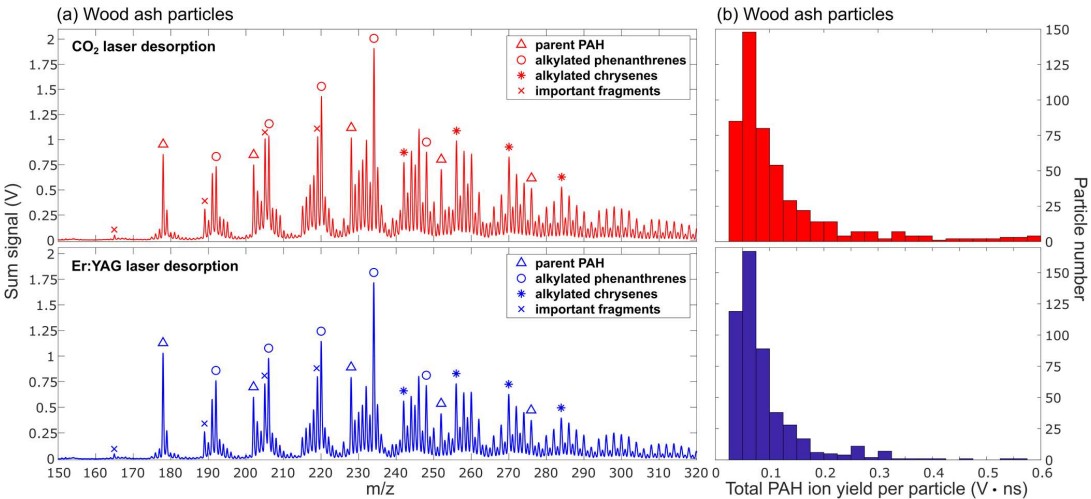

**Figure 3 (a)** Wood ash particles (n = 500) exhibit a very different PAH profile in the LD-REMPI ionization than diesel soot particles, and they show the softwood combustion marker retene (m/z = 234). However, in direct comparison, there are almost no differences between the $CO_2$ laser and the Er:YAG laser for LD. **(b)** The single-particle signal intensity of PAHs again shows a few more particles with strong PAH signals for the $CO_2$ laser.

As a third type of model particle, we focused on tar ball particles. In our case, they were simply sprayed with wood tar and

therefore show a signature of only alkylated phenanthrenes (Fig. 4a), in contrast to a more sophisticated tar ball model that we





analyzed in a previous study using the same LD-REMPI technique (Li et al., 2019). However, they appear to be an appropriate and easier to generate model to study LD for the highly relevant organic aerosols from wood combustion, as they have comparable physical properties such as high viscosity, low volatility, and a brown color with high absorption in the UV-VIS due to their PAH content (Jacobson, 2012; Brege et al., 2021; Li et al., 2019). Also, for this particle type, the mass spectral

differences between the $CO_2$ laser and the Er:YAG laser are negligible. Although, the distribution of PAH signal intensities over the particles (Fig. 4b) reveals a higher number of particles with intense PAH signatures for the Er:YAG laser. The reason for this behavior is not known, but we assume that residues of the solvent methanol led to enhancements in the absorption of the Er:YAG laser, since there is a strong absorption band from the O-H stretching vibration in methanol (Linstrom, 1997). In addition, it was difficult to determine the hit rate in these experiments, because there was a small background of the PAH

signature from particle evaporation effects in the inlet, even when no particle was hit. A particle hit was defined when the sum of the peak areas exceeded 30 mV · ns, based on the distribution of signal intensities per laser shot; see Fig. 4b. This results in a hit rate of 54 % (49 %) for the $CO_2$ laser (Er:YAG laser).

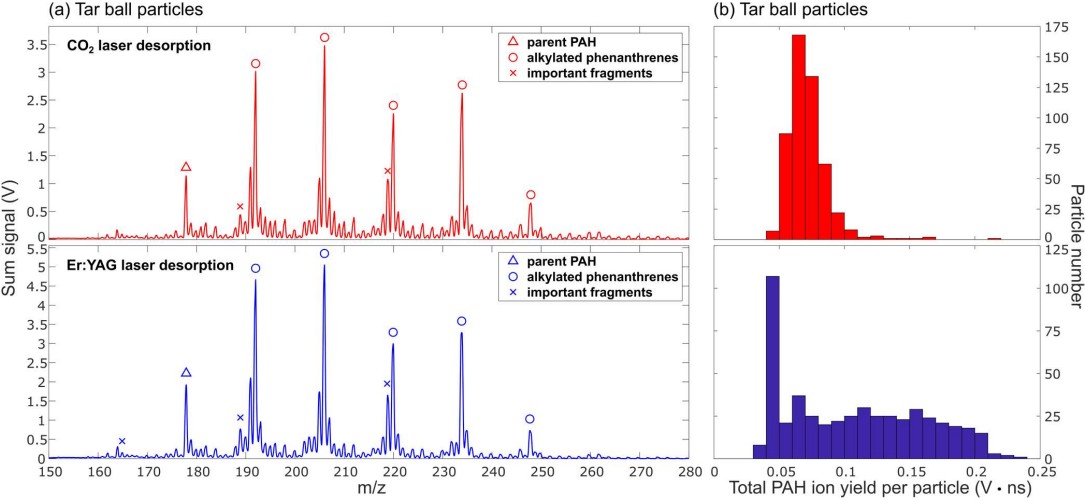

**Figure 4** In our case of sprayed wood tar as a proxy for organic aerosols from wood combustion, the sum PAH spectra (n = 500) show almost exclusively alkylated phenanthrenes. While there are no qualitative differences in the mass spectra between the two lasers used for LD, the Er:YAG laser produced intense PAH spectra more often for this particle type. This can be explained by the strong absorption of the solvent methanol at the 3 μm wavelength of the Er:YAG laser.



### 3.2 Ambient air application of the Er:YAG-based laser desorption

As the laboratory experiments demonstrated the ability of the Er:YAG laser to efficiently desorb organics from different particle types, these results need to be validated in a field study for ambient aerosols. The experiments were conducted at the campus of the University of Eastern Finland in Kuopio, FI (*62.8891° N, 27.6290° E*) from March 1 to 3, 2024. Ambient air masses were sampled at a height of 10 meters above the ground. These air masses came mainly from the south, where the large forests and rural landscapes of Southern Finland and the Baltic States dominate. Backward trajectories were calculated using

the HYSPLIT model with GFS 0.25° meteorological fields as the input file (http://www.ready.noaa.gov/HYSPLIT.php, last access: July 8, 2024, see Fig. S3). The average $PM_{10}$ concentration in this winter time period was 4.0 µg m$^{-3}$ and the average $PM_{2.5}$ concentration was 3.4 µg m$^{-3}$ (weather station Kuopio Niirala, https://en.ilmatieteenlaitos.fi/download-observations, last access: July 8, 2024).

Out of 97,063 optically detected particles in 23 hours, 1,450 revealed PAH spectra, defined by the presence of at least five of

the peaks listed in Tab. 2. The average mass spectra from each 500 PAH-containing particles are compared in Fig. S4 and show a similar overall pattern for both desorption lasers and a comparable hit rate with slightly higher PAH signals and reduced fragmentation for the Er:YAG laser. As mentioned above, size information is only available at the ensemble average (Fig. S1), but not at the single-particle level.

Beyond performance parameters, these experiments on real-world aerosols should reveal whether different wavelengths and

laser parameters affect the results on a single-particle basis. Therefore, the 500 PAH mass spectra from each laser were independently analyzed using the ART-2a clustering algorithm with a vigilance factor of 0.75, 20 iterations, and a learning rate of 0.05 (Sultana et al., 2017). After visual inspection and manual grouping of the 17 (11) ART-2a clusters for the $CO_2$ laser (Er:YAG laser), four main clusters remained, containing ≈ 95 % of the particles; see Fig. 5. The first three clusters in Fig. 5a and 5b show strong signals from parent PAHs, with distribution maxima at low molecular weight PAHs (LMW, cluster

#1), high molecular weight PAHs (HMW, cluster #3), and dominant PAH peaks at m/z = 228 and 252 (cluster #2). Such distributions were previously associated with wood combustion aerosols in ambient air (Passig et al., 2022), and the dominant peaks for benzo[*a*]anthracenes/chrysene (*m/z* = 228) and benzopyrenes (*m/z* = 252) in cluster #2 are an indicator of a higher photochemical age of the particles (Miersch et al., 2019). Of note, not only the mass spectra but also the number of particles in each cluster are comparable for the two desorption lasers, emphasizing a high degree of similarity between the two

desorption processes despite the different wavelengths. The main difference between the two lasers is the higher fragmentation of the $CO_2$ laser, resulting in cluster #4 with dominant fragments, while the fourth cluster produced by the Er:YAG laser again shows parent PAHs as the strongest signals, but with a different intensity distribution.





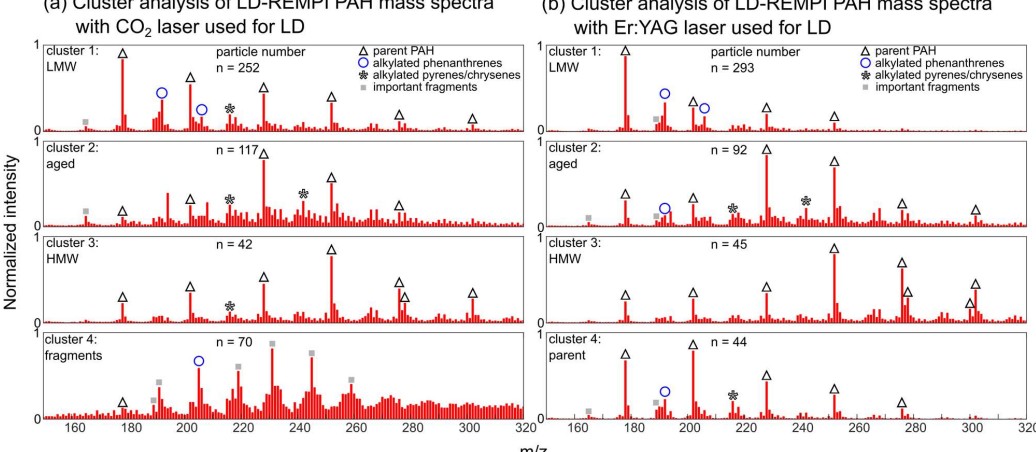

**Figure 5** Main ART-2a clusters of each 500 PAH mass spectra from ambient air particles, exposed to **(a)** the $CO_2$ laser and **(b)** the Er:YAG
laser for LD prior to REMPI photoionization. In the first three clusters, containing > 80 % of the particles, not only the mass spectral
signatures are similar but also the number of particles in the respective cluster. In the fourth cluster (< 14 % of the particles), significant
differences are observed due to the higher fragmentation of the $CO_2$ laser, which can be explained by its beam profile and higher peak
intensity. The results indicate that the compact Er:YAG laser can replace the more commonly used $CO_2$ lasers for single-particle LD also in
ambient air studies.

**3.3 Implementation in the combined ionization scheme for PAHs and inorganics – lab experiments**

Recent developments using spatially and temporally tailored laser pulses allow the combined analysis of PAHs and inorganic
particle constituents by simultaneously inducing REMPI of PAHs in the gaseous plume and LDI of the refractory particle
residue (Schade et al., 2019). We investigated the capability of the Er:YAG laser for its implementation in this ionization
technique for the same laboratory particles and for ambient air. The back reflection mirror was now used to allow the focused

UV beam to be sent back into the ion source for LDI, as shown in Fig. 1b. In addition, the electronics have been modified so
that the $CO_2$ laser and the Er:YAG laser are triggered alternately for LD of the first and second particles as before, and neither
of them is triggered for the third particle. In the latter case, the unfocused UV beam does not induce REMPI because no gas
plume was previously formed, and the back-reflected UV beam only induces LDI, similar to conventional LDI-based SPMS.
This allows a direct comparison of the PAH analysis using the combined ionization scheme with the Er:YAG and $CO_2$ lasers

and with the conventional one-step LDI approach.

Figure 6 shows the results for soot particles, comparing LD with (panel a) the $CO_2$ laser, (panel b) the Er:YAG laser, and
(panel c) one-pulse LDI only. The LDI mass spectra are similar for all three methods, emphasizing that the LD step has no
significant effect on the LDI ion formation. For the PAH spectra in Fig. 6a and 6b, a slightly higher fragmentation can be
noticed compared to the pure LD-REMPI results shown in Fig. 2. This is due to the back-reflected, narrow, high-intensity



beam for LDI that cuts through a portion of the plume and fragments a fraction of the PAH molecules there. Of note, for the

pure LDI process without prior LD, PAH signals are very weak and strongly interfered with molecular fragments (Fig. 6c).

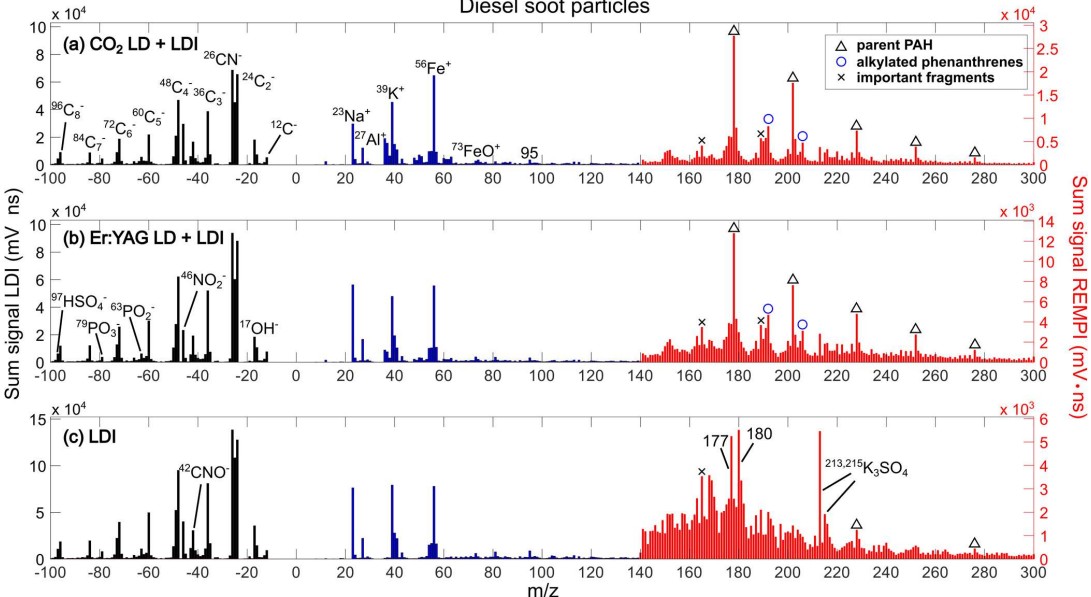

**Figure 6** In the combined LD-REMPI/LDI ionization scheme, PAH detection is combined with conventional inorganic characterization via LDI. Neither the PAH signature nor the LDI-derived composition is significantly affected when the $CO_2$ laser **(a)** is replaced by the Er:YAG laser **(b)** for the LD step. **(c)** In contrast, PAH signatures are barely detectable with conventional single-step LDI ionization. Each n = 500.


For the wood ash particles (Fig. 7), the LDI spectra are comparable, but there are some differences of unknown origin, e.g.,

enhanced phosphate signals when the $CO_2$ laser is used for LD. When comparing the PAH signatures in panels (a) and (b), an

increased signal at m/z = 178 can be recognized for LD with the Er:YAG laser. This trend is also observed for all other PAH

spectra, but to a lesser extent. This may be related to a higher IR absorption of phenanthrene/anthracene at 3 µm compared to

the other PAHs (Laurens et al., 2021). In contrast to the diesel soot experiments, PAHs are now also clearly visible for LDI

only (Fig. 7c). Obviously, the particle matrix of the wood ash particles supports the ionization of PAHs in LDI. These effects

will be discussed in the following sections. Nevertheless, for the combined LD-REMPI/LDI method in Fig. 7a and 7b, the

PAH signals are 5–8 times stronger, compare the Y-axes on the right.

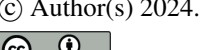



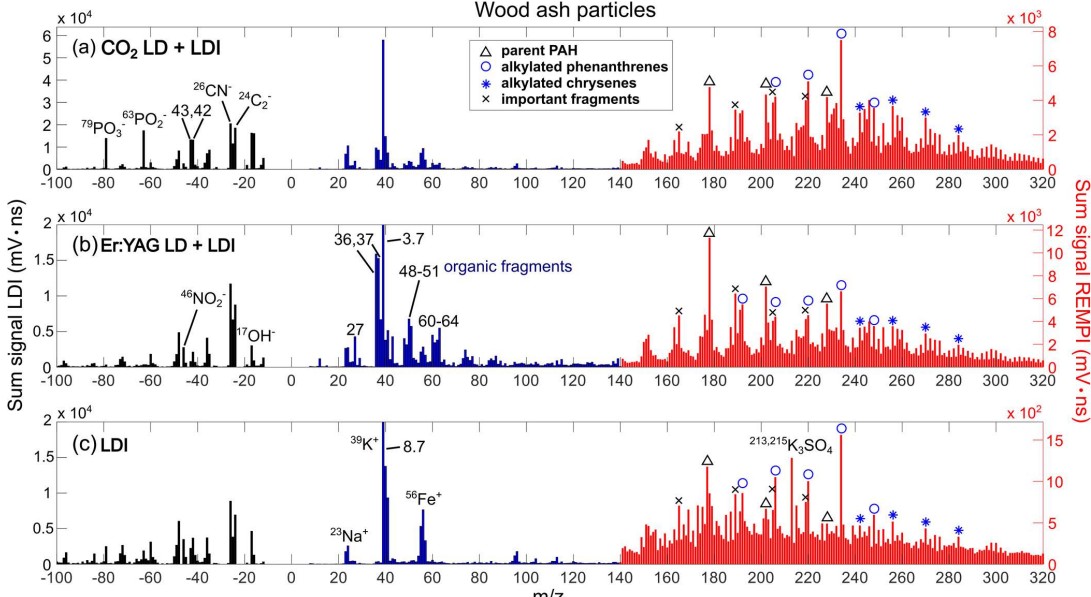

**Figure 7** For wood ash particles, the combined method yields comparable results for **(a)** the $CO_2$ laser and **(b)** the Er:YAG laser applied for LD. Due to the nature of these particles, the single-step LDI method also produces clear PAH signatures, but with a lower sensitivity (compare the Y-axes on the right). Each n = 500.

In the case of tar ball particles, the LDI mass spectra in Fig. 8a and 8b show similar fragmentation patterns. This indicates that resonant excitation of the OH-stretch vibration by the Er:YAG laser does not influence the subsequent LDI when investigating organic aerosols. The PAH signatures are again comparable and approximately four times stronger than the one-step LDI process shown in panel c. Interestingly, the LDI mass spectra in panel c are different, showing stronger signals of soot and inorganic components. This points to the matrix effect of soot: only two types of real-world particles are capable of producing spectra with intact PAHs in one-step LDI: those with strong soot contributions (Zimmermann et al., 2003) and particles with dominant parent PAHs via so-called self-matrix ionization (Zhu et al., 2024). This underlines the superior and more universal PAH detection of two-step methods with LD-REMPI ionization, which is much less dependent on the particle matrix (Woods et al., 2001).



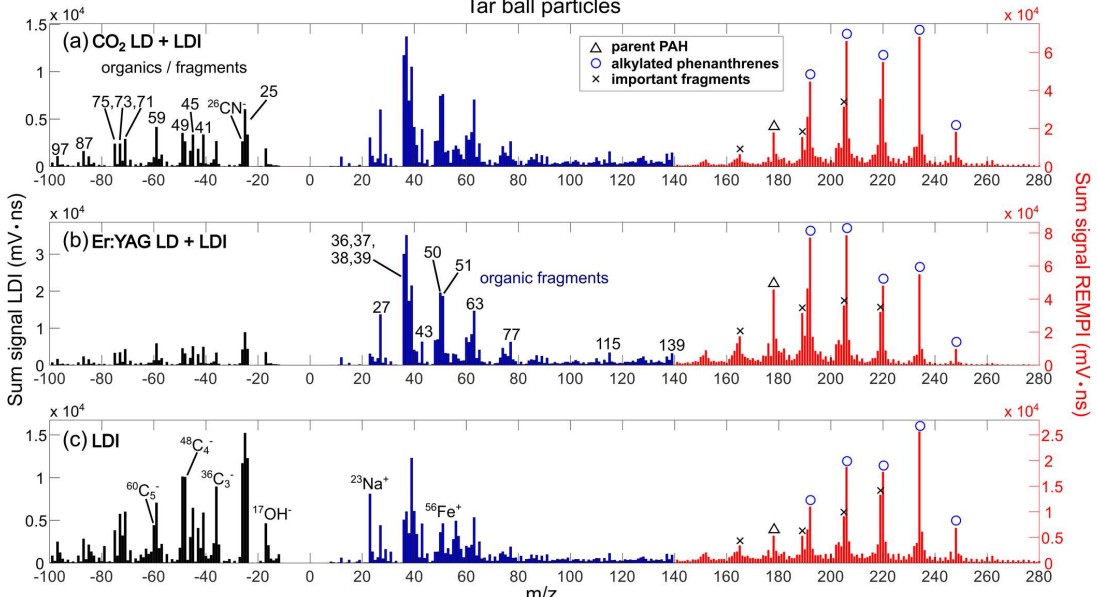

**Figure 8 (a)** and **(b)** Tar ball particles show comparable behavior in the combined LD-REMPI/LDI scheme, independent of the laser used for LD. **(c)** However, in single-step LDI, only particles with soot signatures and more pronounced signals from inorganics produce PAH mass spectra, underscoring the need for an UV-absorbing and ionization-enhancing matrix in LDI. Each n = 500. LDI anion signals are enhanced three times for better visibility.

### 3.4 Implementation in the combined ionization scheme for PAHs and inorganics – ambient air experiments

In an extension of the ambient air experiments shown in Sect. 3.2, we continued the measurements for another 6 hours, now with the combined ionization scheme (Fig. 1b). Data analysis is more complex in this case because the signals from inorganics and PAH patterns should be analyzed separately (Passig et al., 2022). From 30,000 optically detected particles, 16,600 produced a cation LDI spectrum and 12,700 revealed bipolar LDI mass spectra, corresponding to a hit rate of about 55 % (42 % bipolar; see X. Shen et al., 2024 for a comparison of hit rates for different LDI-based SPMS instruments). From these particles, 1,846 showed additional PAH signatures as determined by the presence of at least five of the major PAH peaks listed in Tab. 2 (648 particles for the $CO_2$ laser, 644 for the Er:YAG laser, and 518 for LDI with the Excimer laser only). We selected 500 particles for each laser and analyzed their PAH pattern using ART-2a clustering; see Fig. S5 for the results. A detailed discussion of the particle ensemble is beyond the scope of this paper. In short, the PAH-containing particles were predominantly aged wood combustion particles with strong $K^+$ peaks and pronounced secondary sulfate and nitrate signals, and sometimes soot signatures. Both the Er:YAG laser and the $CO_2$ laser produced comparable patterns of PAH mass spectra,



however, with partly different numbers of particles in the respective classes (Fig. S5). For the LDI process only, without
previous LD and REMPI ionization in the plume, the vast majority of mass spectra were dominated by fragments. This can be
recognized more easily from Fig. 9, where the number of particles with dominant PAH peaks is illustrated for the three different
laser excitation schemes.

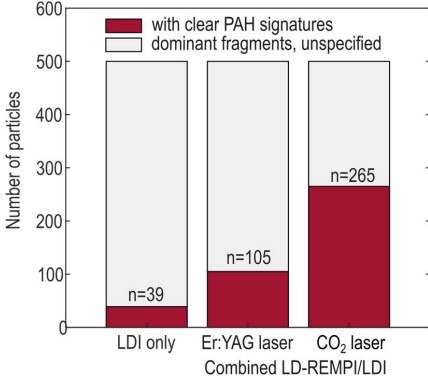

**Figure 9** In a direct comparison of ambient air aerosols, the combined LD-REMPI/LDI ionization schemes yield PAH mass spectra with
much higher efficiency compared to conventional LDI. In the experiments shown here, the prototype Er:YAG laser suffered from pulse-to-
pulse instabilities. For a more stable Er:YAG laser, we expect a similar efficiency in detecting PAHs as for the $CO_2$ laser in the future. The
high prevalence of stable parent PAHs and soot components favors the $CO_2$ laser and LDI ionization compared to the Er:YAG laser. For
some particle types, PAHs are barely detectable with LDI; compare Fig. 6. The mass spectra and a cluster analysis are shown in the
Supplement, Fig. S5.

Obviously, the Er:YAG laser was less efficient in producing clear PAH spectra than the $CO_2$ laser in this experiment. For a
given particle type, achieving efficient plume expansion necessitates a specific laser energy threshold. When pulse energy
fluctuations are significant, as for the prototype laser used here (refer to Supplementary Fig. S2), low pulse energies can result
in inefficient plume expansion, thereby diminishing the hit rate and reducing signal intensities in the mass spectra – a problem
that will be solved with a more stable laser in the future. As noticeable from Fig. 9, the pure LDI process also produced some
clean and evaluable spectra. Once again, it should be noted that these results are strongly dependent on the experimental
conditions and particle type. It is well known that soot can act as an efficient matrix for PAH ionization in LDI, also in SPMS
(Zimmermann et al., 2003). Most of the PAH-containing particles detected here reveal also soot signatures by the presence of
$C_n-$ clusters in negative LDI spectra (Fig. S5), while in other field studies PAHs were mainly detected on organic carbon
particles without clear soot signatures (Passig et al., 2022). Parent PAHs, which are the dominant PAH species in the mass
spectra observed here, are also particularly stable to fragmentation, allowing their analysis with LDI via self-matrix ionization
(Zhu et al., 2024). In many other cases, other PAHs are important, such as alkylated phenanthrenes from incomplete
combustion, which can indicate the particle source (Anders et al., 2023; Anders et al., 2024), or retene, which is a marker for
coniferous wood combustion (Shen et al., 2012b). These PAHs are more susceptible to fragmentation than parent PAHs (Passig



et al., 2021; Gehm et al., 2018; Kruth et al., 2017) and could therefore be underrepresented in LDI. In the present data set,
aromatics other than the parent PAHs are almost absent for all three laser excitation schemes, so a systematic comparison of
the ratios of alkylated/parent PAHs is not possible. In previous tests in Central Europe at warmer temperatures of about 5–
10 °C, only the combined two-step excitation scheme was able to detect PAHs in a similar comparison experiment with LDI
(not shown due to technical difficulties with the SPMS system). In the experiment shown here, the weaker performance of the
Er:YAG laser compared to the $CO_2$ laser is probably a result of the large shot-to-shot variability of its pulse energy (see Fig.
S2). It may also be related to the high abundance of parent PAHs that better resist the higher peak intensities of the $CO_2$ laser
and the LDI laser beam. In general, the combined LD-REMPI/LDI ionization scheme causes more fragmentation than the LD-
REMPI approach alone, since part of the gas plume surrounding the particle is hit by the high-intensity LDI beam. Also,
interferences between PAHs and inorganics occur more frequently, e.g., with $K_3SO_4^+$ at m/z = 213 and 215. However, it has
the advantage of detecting both inorganic components and PAHs of the same particle.

**4 Conclusions and Outlook**

We have directly compared a prototype solid-state laser with a more conventional $CO_2$ laser for laser desorption in two-step
approaches for ionization in SPMS. Although the wavelengths (3 μm Er:YAG solid-state laser vs. 10.6 μm $CO_2$ laser) and
pulse lengths (200 μs Er:YAG laser and 250 ns $CO_2$ laser) differ substantially, the resulting PAH mass spectra show a high
degree of similarity, and the hit rate is also approximately comparable. This can be explained by the desorption mechanism
for laser pulses in the nanosecond to microsecond range, where the laser energy is eventually transferred to thermal energy
and the organic material is ejected by thermal vaporization (Schmidt et al., 2023). Solid-state lasers have multiple advantages
over gas lasers in terms of reliability, usability, and cost. In particular, the regular gas changes required for $CO_2$ lasers and the
necessary provision of gas cylinders are avoided, which is of great importance for measurements in remote areas, on ships, or
in airplanes. This potential can be fully exploited if the excimer laser is also replaced by a (Nd:YAG) solid-state laser, making
the two-step process much more flexible and easier to use. However, the large pulse-to-pulse variability of the current prototype
Er:YAG laser needs to be overcome by developing a more stable system, and the maximum repetition rate should be increased
to allow measurements also in the unsynchronized "free-running" mode for ultrafine particles (Anders et al., 2023; Anders et
al., 2024; Erdmann et al., 2005).

We also evaluated the advantages of the recently developed two-step approach with combined ionization for PAHs and
inorganics (Schade et al., 2019) by direct comparison with conventional LDI ionization. The combined method clearly showed
improvements in the detection of health and environmentally relevant PAHs by spatially separating the LD-REMPI and LDI
processes. The technique facilitates detecting clear PAH signatures and inorganic/refractory particle components
simultaneously. In addition to the higher detection rates for PAHs, we concluded that the combined technique is universally



applicable to mixed aerosols, which has been demonstrated previously (Passig et al., 2022; Anders et al., 2024), whereas
conventional LDI is more limited to soot-containing particles (Zimmermann et al., 2003), pure PAH particles (Zhang et al., 2023), and could potentially introduce a bias toward parent PAHs. Furthermore, we have illustrated that the advantages of two-step processes extend beyond a specific type of desorption laser, as evidenced by comparing a solid-state Er:YAG laser with a $CO_2$-TEA laser.

*Data availability*. Data are available on request from Johannes Passig (johannes.passig@uni-rostock.de).
*Author contributions.* M.S., H.H., and J.P. performed the SPMS experiments. M.S. and J.P. analyzed the data. T.K.B., L.A., A.K., and R.I. provided technical assistance. R.K. and N.G. constructed the Er:YAG laser. M.S. and J.P. wrote the manuscript. J.P. conceived the experiments. J.P. and R.Z. raised funding.
*Conflicts of Interest*. The authors declare no conflict of interest.

**Acknowledgements**

Funded by the Deutsche Forschungsgemeinschaft (DFG, German Research Foundation) - SFB 1477 "Light-Matter Interactions at Interfaces", project number 441234705. This research was supported by the Federal Ministry for Education and Research as part of the joint project HazarDust under the funding code 13N15569.
The authors would like to thank Olli Sippula and colleagues at the University of Eastern Finland for hosting the ambient air
experiments.

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
