# Peer review of "A solid-state IR laser for two-step desorption/ionization processes in single-particle mass spectrometry"

_EGUsphere, 2024_

## Referee Comment (RC2)

This manuscript presents an exploration into technical developments in single particle mass spectrometry (SMPS) aimed to improve the detection and identification of various PAHs in individual aerosol particles. From the onset the authors note that to generate high-quality single particle mass spectra of particle-bound PAHs requires evaporating/desorbing the PAHs and ionizing them in the gas phase by REMPI. Indeed, two-laser schemes using a $CO_2$ laser in the IR to evaporate semi-volatile molecules, followed by UV ionization have previously been shown to yield significant improvements in SPMS. In addition to reducing fragmentation and matrix effects, it also was shown to result in highly reproducible mass spectra with significantly smaller particle-to-particle fluctuations (e.g. (Zelenyuk, Yang et al. 2009)).

The central issue here is to explore the use of a solid state Er:YAG laser producing light at ~3 μm as an alternative to the $CO_2$ laser. However, while the authors emphasize the cost-effectiveness and compact size of the Er:YAG laser, no details on its size, weight, cost, and ease of operation are provided. The discussion of the Er:YAG laser features and performance should be extended prior to manuscript publication.

It is important to note that aside from the wavelength, there are other important differences between the $CO_2$ and Er:YAG that are clearly listed in Table 1. Most notably, the Er:YAG pulse length is 200 μs and the laser beam, as used, is 5x5 mm, as compared to 250 ns pulse width and a 1x1 mm beam size for the $CO_2$ laser. This translates to a difference in irradiance of a factor of 5,000!

A note on IR wavelengths used here. As it turns out the absorption by PAHs at both the Er:YAG, at 3 μm and $CO_2$, at 10.6 μm, are on the edges of the IR absorption bands of PAHs. This brings out the question of the role of differences in IR absorption cross-section of the various PAHs in determining their absolute and relative abundance using the observed mass spectral peak intensities. It is rather surprising to find that despite the large differences between the two IR light sources, the mass spectra generated using $CO_2$ and Er:YAG presented in the manuscript, are nearly the same.

To assure gentle REMPI, the evaporated gas plume is ionized using a 4 mJ/pulse KrF excimer laser pulse (248 nm), operated with a 5x10 mm laser beam. As a result, the LD mass spectra of particles used in this study are generally of high quality and relatively easy to assign.

The manuscript presents a series of measurements on what the authors call "laboratory particles": diesel soot, wood ash, and tar balls particles, which all exhibit nearly IR laser independent mass spectra. It is most interesting that for most of the mass spectra, the absolute and relative peak intensities are independent of desorption laser. Why, I am not clear. Could the authors discuss the possible reasons for these observations? Are both IR lasers evaporate the whole particle over the entire particle size range? Given the significant pulse-to-pulse energy variations reported in Figure S2 for the Er:YAG laser, are particles completely evaporated at lower pulse energies? If so, the histograms of the total PAH ion yield per particle for the Er:YAG laser, shown in Figures 2 and 3, seems to be inconsistent with the histogram of the pulse energies presented in Figure S2. The authors state that *"The total reduction in hit rate resulting from the combined loss in duty cycle and*

*pulse-to-pulse variations is estimated to be ≈ 30–50 %, compared to a hypothetical stable Er:YAG laser to be developed for irregular triggering in SPMS.",* which seems to suggest that lower pulse energies are insufficient to generate *"a sufficient plume".* The authors also state that *"the pulse energy required to generate a sufficient plume is highly dependent on the particle properties and composition."* Do these particle properties include particle size? I expect that both, particle size and laser power, can significantly affect the observed mass spectra and apparent hit rate, but were not investigated in the present study.

It also important to note that the reported hit rates for wood ash particles were ~ an order of magnitude lower compared to those observed for diesel soot 2% (4%) vs. 38% (49%) for the $CO_2$ and (Er:YAG) lasers. The authors state that *"This is due to the nature of the sample, which contains many burnt ash particles and fewer OC/soot particles containing PAHs".* Ash particles should have been efficiently detected and identified by the LDI. Was the observed hit rate for the data presented in Figure 6 a factor of ~10 higher compared to those in Figure 3? Moreover, the mass spectra of ash-rich particles would be very different compared to the PAH-containing particles. Were different types of the mass spectra observed in the cases where the LDI was used?

The experiments on ambient particles are clearly more challenging. Despite this fact, the results on PAH-containing particles for the two IR lasers are again very similar. This set of experiments clearly demonstrates that the solid state Er:YAG can be used to evaporate semi-volatile molecules from particles, much like the $CO_2$ laser, which is the central point of the paper.

Characterization of mixed particles, containing semi-volatiles and nonvolatile species, requires a more intense UV laser to ablate the nonvolatile fraction in the particle. It is accomplished by using a reflector that focuses the Excimer laser pulse, such that each particle is hit with the IR laser first, then the evaporated plume is ionized with unfocused excimer laser, that same beam is reflected and focused to hit what remains in the particle for LDI.

The authors apply this scheme to all three laboratory particle samples, presenting the data for the two of the IR lasers, including the LDI step, as well as the mass spectra generated by the excimer laser only (LDI). The first two examples provide a clear demonstration of how much better the mass spectra are when IR laser is used. Surprisingly, the LDI mass spectra for tar ball particles are nearly the same as those with IR evaporation, which the authors explain by the presence of soot in these particles. If so, why the LDI mass signatures of soot are "missing" in panels (a) and (b) of Figure 8?

The ambient particles experiment also show improved mass spectral signatures with IR evaporation. At present, the Er:YAG is clearly less efficient than the $CO_2$ laser. However, as the authors note, future developments of the Er:YAG could result in significant improvements.

The weakest aspect of this manuscript relates to particle size. Particle evaporation and the amount of material in the plume are expected to be strongly dependent on particle size and hence affect the results of the entire study. The size distributions (aerodynamic diameters in the free molecular regime?) are shown in the supplement. All these size distributions show particles that

are significantly larger than what I would expect for "real" diesel soot, wood ash, and ambient particles. I assume this is due to the particle generation (dispersion of the collected and milled bulk samples) and aerosol concentration methods used.

The manuscript notes that "the instrument in this configuration could not record individual particle size information but only average size distributions." and, "size information is only available at the ensemble average," which I find confusing. You must detect each particle twice and "a hit" means it was properly detected (i.e. it was not a false detection or free-running laser) does that not yield a size for each particle? As I mentioned above, the size-dependent information would be critical to understand the performance of the new Er:YAG laser, which is/should be a focus of this paper.

Overall, the paper is well-written and is suitable for publication in AMT after revision.

Zelenyuk, A., J. Yang and D. Imre (2009). "Comparison between mass spectra of individual organic particles generated by UV laser ablation and in the IR/UV two-step mode." International journal of mass spectrometry **282**(1): 6-12.

---

## Author Comment (AC1)

Referee comments and author response for egusphere-2024-2587

**A solid-state IR laser for two-step desorption/ionization processes in single-particle mass spectrometer**

Note:
Reviewer comments are in normal format.
*Author responses are in italics, blue.*
**Changes** that were made to the manuscript are in **bold** face.

**Referee 1**

**General comments:**

The authors discuss the advantages and disadvantages of using a new type of IR-laser for laser desorption followed by REMPI and LDI by an UV laser. The use of a solid-state Er:YAG instead of the widely used CO2 laser offers a more compact design with less maintenance required. However the prototype character of the Er:YAG still leads to instabilities in laser energy and less reproducibility.

The authors show on a variety of aerosol species, that the new laser leads to reasonable and qualitatively good spectra with similar quality compared to the established method, still pointing out the existing problems and differences.

The comparison of the LD/REMPI on three species and LD/REMPI+LDI on the same three species seems very repetitive and does not provide the necessary insight justifying for 6 plots. I suggest, reducing this part to 3 plots, shifting Fig. 2,4 and 6 along with its corresponding text into the supplement. In the main article a short note to the results of Fig. 2,4, and 6 with reference to the supplement should suffice.

*We agree with the reviewer on the length issues.* **As suggested by the reviewer, we have moved three figures with mass spectra including the corresponding text to the Supplement and added short notes on the respective results in the manuscript.**

Instead the description of the Er:YAG laser falls short, since it is the main news of the publication. Please consider adding more info, e.g. about the dimensions, pulse shape and pumping.

**We added the required information in a new section.**

All-in-all the writing and the structure of the paper is very clear, enabling a good understanding of the topic and what the authors are trying to get across. The supplement is useful and offers a reasonable amount of extra information.
I suggest publication in AMT after some minor to major revision.

*We thank the reviewer for his/her time and positive feedback about the manuscript.*

**Specific comments:**

Consider shifting some figures and text for the spectral comparisons into the supplement (eg. Fig. 2,4, and 6)

**Figure 3, 4 and 7 were shifted to the supplement, the text was also shortened in the main article and a full explanation can be found in the supplement now.**

Consider a more detailed description of the Er:YAG laser, e.g. about the dimensions, pulse shape and pumping.

*We added a section about Er:YAG laser including a detailed description in the methods section of the paper.*

The spectra are compared in detail, however, it is not clear, whether minor differences state a problem. Please state at some point, that due to the missing quantitativeness of the measurement method, minor differences in the acquired spectra do not undermine the applicability of the new laser.

*We added a corresponding statement early in the results section 3.1: "It is also important to note that in SPMS, due to the limits in quantitative measurement, minor differences in the mass spectra obtained do not undermine the applicability of the Er:YAG laser."*

line 100: "(Zhuo, Z., Su, B., Xie, Q., Li, L., Huang, Z., Zhou, Z., Mai, 2021)."
This should be abbreviated as Zhuo, Z. et al., 2021. Name of Mai is missing the Z. and last author Tan, G. is not mentioned, rework the reference.

*Thank you! Corrected.*

line 108f: "a 14-bit digitizer card (ADQ14…"
To my knowledge the ADQ14 provides 4 channels. If this instrument uses all four channels, similar to PALMS-NG, ALABAMA, ERICA and others, clarify here: "2 channels per polarity"

*The ADQ14 card is available with 1,2 or 4 channels. We use the 2-channel version, with only one channel per polarity and added a respective statement in the manuscript.*

line 124: "This is due to the lack of Q-switching. Within the 200 µs pulse duration,"

It is not 100% clear, that the lack of Q-switching and the pulse duration both refer to the Er:YAG laser. Please rephrase.

*Thank you. Corrected: "This is due to the lack of Q-switching for the Er:YAG laser. Within the 200 µs pulse duration,…"*

line 129: ", see Fig. S2)"
This can be more specific, like "a histogram of laser energies can be found in Fig. S2)"

*Ok. Done. "(up to ± 50 % calculated on the average pulse energy, a histogram of pulse energies can be found in Fig. S2)."*

line 144: "no LD"
strictly LDI is a form of LD, thus rather write "no prior LD"

*Thank you. Corrected accordingly.*

line 153: "was not exposed to LD"
compare comment above, rather write "was not exposed to the IR LD laser"

*Corrected.*

Fig. 1
The distance between LD and REMPI/LDI seems pretty large. This is totally ok for a sketch, but somewhere it should be clarified, how small the distance is in reality.
*The distances differ slightly between the Er:YAG and CO₂ lasers due to the broad beam of the Er:YAG laser, which spatially overlaps with the unfocused UV beam used for REMPI. To clarify the dimensions without overcomplicating the description, we included two brief notes at the beginning of the respective sentences, as*

*follows: "**In the ion source,** the two desorption lasers, i.e., the "standard" CO2 laser and the experimental Er:YAG laser, are alternately fired at the respective particles, generating a small gas plume for each particle. **Few µs later,** the unfocused UV laser pulse ionizes the PAHs in the expanding plume via REMPI. "*

line 251: "resulting in cluster #4 with dominant fragments"
I am not convinced by the interpretation of a very small peak at ~178 as the only parent molecule and everything else as fragments. All other spectra show more parent molecules than fragments. Can the peaks not be explained by a different parent molecule, possibly hydrogenated or methylated PAH?

*Thank you for the comment and discussion! The mass spectrum of cluster #4 can be identified as a typical fragmentation pattern for several reasons. Oxidized PAHs exhibit a significantly lower REMPI cross section and therefore typically do not contribute to the major peaks. Methylated PAHs, on the other hand, have even m/z numbers (e.g., 192, 206, 220, etc.). The strongest peaks in this spectrum, however, display odd m/z numbers, which is characteristic of PAH fragments (e.g., resulting from H-atom abstraction). Although a detailed discussion of this is beyond the scope of this paper, we have added a note regarding the odd m/z values to the text.: "The primary difference between the two lasers is the higher fragmentation caused by the $CO_2$ laser, which leads to the formation of cluster #4, characterized by dominant fragments **with odd m/z values."***

line 258f: "The results indicate that the compact Er:YAG laser can replace the more commonly used CO2 lasers for single-particle LD also in ambient air studies."
This conclusion is not part of the graph description and should only be mentioned in the plain text, not in the caption.

*Yes, we agree. Accordingly, **we removed the conclusion from the caption and added into the text.***

line 263f: "We investigated the capability of the Er:YAG laser for its implementation in this ionization technique"
This sounds like the Er:YAG does the REMPI/LDI, which is not the case. Rephrase like "We investigated the capability of LD by the Er:YAG laser in combination with this ionization technique"

*Thank you! **Changed accordingly.***

line 281f: "but there are some differences of unknown origin, e.g., enhanced phosphate signals when the CO2 laser is used for LD."
In the LD scheme without LDI, only cations have been investigated. Do the wood ash LD/Rempi spectra show the phosphate signal in the anions as well? Since only the LD laser is changed between 7a and 7b, a difference in the LDI spectrum would indeed be astonishing.
*No mass spectrometric peaks were visible for anions using LD-REMPI without LDI. We believe that the relatively short and intense CO2 laser pulse excites the particle in such a way as to enhance ion formation of some particle components in the subsequent LDI process, possibly by selectively vaporizing different materials. In contrast, the long and low intensity pulse of the Er:YAG laser results in a slow heating of the whole particle. However, the data presented here do not allow a sophisticated investigation of the underlying processes.*

line 320: "the vast majority of mass spectra were dominated by fragments"
At this point I am not sure, whether I know what you mean. Since you differentiate between LDI spectrum and PAH spectrum within the same spectrum and you use the term fragment mainly in the context of PAHs, I would guess you mean most of the PAH spectra (mz>140) were dominated by fragments.
*Yes, exactly. We added this information to the sentence: "For the LDI process only, without previous LD and REMPI ionization in the plume, **the vast majority of the PAH mass spectra** were dominated by fragments."*

**Technical comments:**

line 161: "isobar substances"
I think the correct term is isomeric
*Sorry, of course. Corrected.*

line 161f
several parts of the sentence are written in italic for no obvious reason
*Corrected.*

line 162f
replace above and below by upper panel and lower panel or similar (top panel, bottom panel)
*Thank you. Corrected.*

Table 2
Some lines are not separated by an empty line, in some lines the m/z are not aligned with the species
*Corrected and aligned.*

line 181: "of each 500 diesel soot particles"
The expression sounds unfamiliar, shouldn't it be "of 500 diesel soot particles, each"
*Corrected.*

line 235: "from each 500 PAH-containing particles"
What does each 500 mean in the context of 1450 measured spectra? Should it be "from a representative set of 500 PAH-containing particles, each"
*Yes, thank you. Corrected and clarified.*

line 280, line 292, line 305: "Each n = 500."
n=500 each.
*Corrected.*

line 370: "LDI ionization"
This is a pleonasm. Rather write LDI method or LDI scheme.
*Corrected to "…**LDI method**…".*

line 536f: "…Con- centrator… Spectrome- try…"
Please check spelling
*Corrected.*

line 543
The title of the reference is in capitals, not in agreement with the other references
*Capitalization seems to be common for patent titles. We would leave the decision to the editorial team.*

---

## Author Comment (AC2)

Referee comments and author response for egusphere-2024-2587

**A solid-state IR laser for two-step desorption/ionization processes in single-particle mass spectrometer**

Note:
Reviewer comments are in normal format.
*Author responses are in italics, blue.*
**Changes** that were made to the manuscript are in **bold** face.

**Referee 2**

This manuscript presents an exploration into technical developments in single particle mass spectrometry (SMPS) aimed to improve the detection and identification of various PAHs in individual aerosol particles. From the onset the authors note that to generate high-quality single particle mass spectra of particle-bound PAHs requires evaporating/desorbing the PAHs and ionizing them in the gas phase by REMPI. Indeed, two-laser schemes using a CO2 laser in the IR to evaporate semi-volatile molecules, followed by UV ionization have previously been shown to yield significant improvements in SPMS. In addition to reducing fragmentation and matrix effects, it also was shown to result in highly reproducible mass spectra with significantly smaller particle-to-particle fluctuations (e.g. (Zelenyuk, Yang et al. 2009)).

*We thank the referee for his/her detailed review and the positive feedback!* **We added the suggested reference to our manuscript.**

The central issue here is to explore the use of a solid state Er:YAG laser producing light at ~3 μm as an alternative to the CO2 laser. However, while the authors emphasize the cost-effectiveness and compact size of the Er:YAG laser, no details on its size, weight, cost, and ease of operation are provided. The discussion of the Er:YAG laser features and performance should be extended prior to manuscript publication.

*We agree with the reviewers opinion and* **added a section about Er:YAG laser including a detailed description.**

It is important to note that aside from the wavelength, there are other important differences between the CO2 and Er:YAG that are clearly listed in Table 1. Most notably, the Er:YAG pulse length is 200 μs and the laser beam, as used, is 5x5 mm, as compared to 250 ns pulse width and a 1x1 mm beam size for the CO2 laser. This translates to a difference in irradiance of a factor of 5,000!
*Exactly! These huge differences in laser intensity and pulse length (but not for the pulse energy a particle is exposed to) are stated several times throughout the manuscript, e.g. in the abstract and in the setup description section 2.3. An explanation for the observed similarity of the resulting spectra is e.g. given early in the results section 3.1: "It is also remarkable, that the signal intensities for the summed mass spectra of the same number of particles are comparable for both lasers, despite the very different laser intensities, durations, and wavelengths. The high degree of similarity can be explained by the underlying physical mechanisms of LD, where the energy transferred to the particle eventually results in thermal desorption for laser pulses longer than a few picoseconds…". The reason for the similarity is the rapid energy transfer from the electrons to the lattice (picoseconds) that results in a pure thermal vaporization of the non-refractory substances, nearly independent of the length of the exciting laser pulse.*

A note on IR wavelengths used here. As it turns out the absorption by PAHs at both the Er:YAG,

at 3 μm and CO2 , at 10.6 μm, are on the edges of the IR absorption bands of PAHs. This brings out the question of the role of differences in IR absorption cross-section of the various PAHs in determining their absolute and relative abundance using the observed mass spectral peak intensities. It is rather surprising to find that despite the large differences between the two IR light sources, the mass spectra generated using CO2 and Er:YAG presented in the manuscript, are nearly the same.

*We sincerely thank the reviewer for this valuable comment. The only consistent difference in the mass spectra is a slightly higher relative intensity of the peak at m/z = 178 when using the Er:YAG laser.* **We added a statement in the discussion of mass spectral differences in section 3.3: "The wavelengths of the Er:YAG laser (3 μm) and the CO₂ laser (10.6 μm) lie at the respective edges of the PAH's IR absorption band, which could contribute to subtle differences in the mass spectral patterns."**

To assure gentle REMPI, the evaporated gas plume is ionized using a 4 mJ/pulse KrF excimer laser pulse (248 nm), operated with a 5x10 mm laser beam. As a result, the LD mass spectra of particles used in this study are generally of high quality and relatively easy to assign.
*We thank the reviewer for the positive comment on our method.*

The manuscript presents a series of measurements on what the authors call "laboratory particles": diesel soot, wood ash, and tar balls particles, which all exhibit nearly IR laser independent mass spectra. It is most interesting that for most of the mass spectra, the absolute and relative peak intensities are independent of desorption laser. Why, I am not clear. Could the authors discuss the possible reasons for these observations? Are both IR lasers evaporate the whole particle over the entire particle size range? Given the significant pulse-to-pulse energy variations reported in Figure S2 for the Er:YAG laser, are particles completely evaporated at lower pulse energies? If so, the histograms of the total PAH ion yield per particle for the Er:YAG laser, shown in Figures 2 and 3, seems to be inconsistent with the histogram of the pulse energies presented in Figure S2.

*The experimental data does not provide direct information about the degree of evaporation for the respective particles. Both lasers were optimized for their parameters, including laser energy, timing cycle, and spot size. For both IR lasers, some particles are effectively desorbed while others are not. The absolute peak intensities, however, are not independent of the desorption laser, as shown in (new number) Figure 4, where the Er:YAG laser generates approximately four times stronger PAH signals in the sum spectra. Despite this, the relative peak intensities remain very similar, which is a key finding of this study and highlights the applicability of the Er:YAG laser. We attribute this behavior to the LD mechanisms, which ultimately lead to thermal evaporation for both laser types, as discussed in Sections 3.1 and 3.3.*

*We thank the reviewer for pointing out a potential contradiction in the signal distribution Fig. 2 vs. Fig. S2) under the assumption of complete particle evaporation. Indeed, we don't believe that particles are completely evaporated. We extended the discussion about the similarity by adding the following statements: "**In addition, saturation effects in the REMPI process might play a role (Gehm et al., 2018). The extent of evaporation during LD cannot be determined from the experimental data. However, the broad size distribution (Fig. S1) and the high pulse-to-pulse variability (Fig. S2) suggest that complete vaporization of PAHs from the particles is unlikely**."*

The authors state that "The total reduction in hit rate resulting from the combined loss in duty cycle and pulse-to-pulse variations is estimated to be ≈ 30–50 %, compared to a hypothetical stable Er:YAG laser to be developed for irregular triggering in SPMS.", which seems to suggest that lower pulse energies are insufficient to generate "a sufficient plume". The authors also state that "the pulse energy required to generate a sufficient plume is highly dependent on the particle properties and

composition." Do these particle properties include particle size? I expect that both, particle size and laser power, can significantly affect the observed mass spectra and apparent hit rate, but were not investigated in the present study.

*We agree with the reviewer that the particle size is an important factor, while it is also reasonable that larger particles are not completely evaporated which might mitigate such differences to some extent. However, the instrumental setup used for these experiments differs from its standard configuration, where individual particle size data is available. The software, which normally combines individual particle size data with the corresponding mass spectra, was modified to enable alternate triggering of the lasers. As a result, particle size data is only available for the entire ensemble (Fig. S1), and it was not possible to assign size data to individual particles. Consequently, the effect of particle size could not be studied in detail.*

It also important to note that the reported hit rates for wood ash particles were ~ an order of magnitude lower compared to those observed for diesel soot 2% (4%) vs. 38% (49%) for the $CO_2$ and (Er:YAG) lasers. The authors state that "This is due to the nature of the sample, which contains many burnt ash particles and fewer OC/soot particles containing PAHs". Ash particles should have been efficiently detected and identified by the LDI. Was the observed hit rate for the data presented in Figure 6 a factor of ~10 higher compared to those in Figure 3? Moreover, the mass spectra of ash-rich particles would be very different compared to the PAH-containing particles. Were different types of the mass spectra observed in the cases where the LDI was used?

*Yes, LDI spectra were generated for the majority of optically detected wood ash particles (hit rate >50%), but only 2–4% of the wood ash particles exhibited PAH signatures. The hit rate and histogram data consistently refer to the PAH mass spectra, as the LD(-REMPI) process is the main focus of this paper. The LDI spectra shown in Fig. 6 (now Fig. S7) represent the sum signal of the same 500 particles that produced the sum REMPI signals shown on the right. In a single-particle analysis, different particle types within the wood ash can be distinguished, with potassium-rich particles showing low OC signals and PAH-rich particles also displaying OC fragments in the LDI portion of the spectrum. However, such detailed single-particle analyses seem beyond the scope of this paper.*

The experiments on ambient particles are clearly more challenging. Despite this fact, the results on PAH-containing particles for the two IR lasers are again very similar. This set of experiments clearly demonstrates that the solid state Er:YAG can be used to evaporate semi-volatile molecules from particles, much like the $CO_2$ laser, which is the central point of the paper. Characterization of mixed particles, containing semi-volatiles and nonvolatile species, requires a more intense UV laser to ablate the nonvolatile fraction in the particle. It is accomplished by using a reflector that focuses the Excimer laser pulse, such that each particle is hit with the IR laser first, then the evaporated plume is ionized with unfocused excimer laser, that same beam is reflected and focused to hit what remains in the particle for LDI.
*Thank you. Yes, this is the working principle of the method.*

The authors apply this scheme to all three laboratory particle samples, presenting the data for the two of the IR lasers, including the LDI step, as well as the mass spectra generated by the excimer laser only (LDI). The first two examples provide a clear demonstration of how much better the mass spectra are when IR laser is used. Surprisingly, the LDI mass spectra for tar ball particles are nearly the same as those with IR evaporation, which the authors explain by the presence of soot in these particles. If so, why the LDI mass signatures of soot are "missing" in panels (a) and (b) of Figure 8?

*The LDI mass signatures of soot are visible in panels (a) and (b) of Fig. 8 (now Fig. 5). However, their relative intensity is lower compared to the LDI-only data shown in panel (c), making them somewhat challenging to annotate. In the single-step LDI process, soot components facilitate the ionization of PAHs. As a result, the mass spectra in panel (c) highlight the bias of the LDI method toward soot-containing particles, leading to an enhanced representation of EC signatures. We modified the caption of Fig. 8 (now Fig. 5) improve clarity here: "**However, in single-step LDI, only particles with soot signatures and more pronounced signals from inorganics produce PAH mass spectra, underscoring the need for an UV-absorbing and ionization-enhancing matrix in LDI. This is reflected by the higher relative ion signals of soot clusters in panel (c).**"*

The ambient particles experiment also show improved mass spectral signatures with IR evaporation. At present, the Er:YAG is clearly less efficient than the CO2 laser. However, as the authors note, future developments of the Er:YAG could result in significant improvements.
The weakest aspect of this manuscript relates to particle size. Particle evaporation and the amount of material in the plume are expected to be strongly dependent on particle size and hence affect the results of the entire study. The size distributions (aerodynamic diameters in the free molecular regime?)

*We agree with the referee regarding the lack of a systematic size-dependent investigation. As previously mentioned, this was not feasible because the instrument had to be modified to enable the alternate firing of the different lasers. In this configuration, size information was recorded, but it could not be linked to individual particles. However, this "alternate firing" setup allowed us to measure the same aerosol and even ambient air simultaneously, ensuring that the only variable was the LD laser itself. It is important to note that the particle ensemble, including the size distribution, was identical when comparing the lasers for LD. Therefore, if significant differences in size-resolved behavior existed between the lasers, these differences should be evident in the plots.*

All these size distributions show particles that are significantly larger than what I would expect for "real" diesel soot, wood ash, and ambient particles. I assume this is due to the particle generation (dispersion of the collected and milled bulk samples) and aerosol concentration methods used.

*Yes, laboratory particles were collected and re-dispersed using a small-scale powder disperser (section 2.1). Therefore, many particles are relatively large agglomerates.*

The manuscript notes that "the instrument in this configuration could not record individual particle size information but only average size distributions." and, "size information is only available at the ensemble average," which I find confusing. You must detect each particle twice and "a hit" means it was properly detected (i.e. it was not a false detection or free-running laser) does that not yield a size for each particle? As I mentioned above, the size-dependent information would be critical to understand the performance of the new Er:YAG laser, which is/should be a focus of this paper.

*As mentioned before, the experiments were conducted in a modified configuration to enable the alternate firing of the lasers. Every particle was detected twice and the hit rate as well as size distribution was calculated on this basis, but the assignment between individual particle size data and mass spectral data was not possible in this experiment. We modified the description of the triggering setup as follow (section 2.3): "**In this configuration, each particle was detected twice, and size data was recorded along with mass spectra as in normal operation. However, the individual particle size data could not be directly linked to individual spectra, meaning only average size distributions could be obtained. This "alternate firing" setup allowed for simultaneous measurement of the same aerosol and even ambient air, ensuring that the only variable was the LD laser.**" Since the particle ensemble and size distribution was exactly the same for each laser, we can exclude a strong size-dependent difference for the two lasers.*

Overall, the paper is well-written and is suitable for publication in AMT after revision.

*We thank the reviewer for taking his time, the detailed review and his/her positive feedback.*

Zelenyuk, A., J. Yang and D. Imre (2009). "Comparison between mass spectra of individual organic particles generated by UV laser ablation and in the IR/UV two-step mode." International journal of mass spectrometry 282(1): 6-12.

***Reference added in the manuscript.***

---

## Referee Report (RR1)

Referee comments for the revised version of egusphere-2024-2587

A solid-state IR laser for two-step desorption/ionization processes in single-particle mass spectrometry

General comments:

The authors discuss the advantages and disadvantages of using a new type of IR-laser for laser desorption followed by REMPI and LDI by an UV laser on the example of various aerosol species.

The authors have considered all suggested revisions and made valuable changes to the manuscript. The publication is in a good shape, the addressed topics are appropriate with regard of the scientific importance and the structure and language are well understandable.

The supplementary information is of good quality.

I suggest publication in AMT after very few minor changes.

Specific comment:

line 22ff: "Additionally, we compared the novel two-step ionization scheme for the combined detection of aromatic molecules and inorganics with conventional single-step laser desorption/ionization (LDI) for the detection of polycyclic aromatic hydrocarbons (PAH) in laboratory and field experiments"

It is not completely clear to me, what this sentence should say. To me it sounds like you compared LD/REMPI for aromatics and inorganics with LDI only for PAH. Shouldn't it say:

We compared a novel two-step ionization (LD-REMPI/LDI) with the conventional single-step LDI regarding the potential to detect PAHs and inorganics in laboratory and field experiments.

Technical comments:

line 181: "each 500 diesel soot particles"

500 diesel soot particles each

line 232 and 234:

access the webpages again and change date of last access

line 258: "each 500 PAH mass spectra from ambient air particles"

500 PAH mass spectra from ambient air particles each

Table 2

Some lines are not separated by an empty line, in some lines the m/z are not aligned with the species. This has not been changed since the former version

Caption of Supplementary figure S1: "each n = 1200"

n=1200 each

Figure S5:

The scale of the Meters AGL can be zoomed by about a factor of 2

Supplement line 34: "as discussed before"

as discussed in section 3.3

---

## Author Response (AR2)

Dear Editor,

thank you for handling our manuscript!

We would like to thank the reviewer for his/her work. We considered all revisions and technical corrections suggested by the reviewer. The only issue we couldn't correct is to zoom the Y-axis of the pdf plots of the HYSPLIT trajectory heights in the Supplement figure S5.

Thank you and best regards,

Johannes Passig

**Referee comments for the revised version of egusphere-2024-2587**

A solid-state IR laser for two-step desorption/ionization processes in single-particle mass spectrometry

**General comments:**

The authors discuss the advantages and disadvantages of using a new type of IR-laser for laser desorption followed by REMPI and LDI by an UV laser on the example of various aerosol species. The authors have considered all suggested revisions and made valuable changes to the manuscript. The publication is in a good shape, the addressed topics are appropriate with regard of the scientific importance and the structure and language are well understandable.

The supplementary information is of good quality.

I suggest publication in AMT after very few minor changes.

**Specific comment:**

line 22ff: "Additionally, we compared the novel two-step ionization scheme for the combined detection of aromatic molecules and inorganics with conventional single-step laser desorption/ionization (LDI) for the detection of polycyclic aromatic hydrocarbons (PAH) in laboratory and field experiments"

It is not completely clear to me, what this sentence should say. To me it sounds like you compared LD/REMPI for aromatics and inorganics with LDI only for PAH. Shouldn't it say: We compared a novel two-step ionization (LD-REMPI/LDI) with the conventional single-step LDI regarding the potential to detect PAHs and inorganics in laboratory and field experiments.

**Technical comments:**

line 181: "each 500 diesel soot particles"

500 diesel soot particles each

line 232 and 234:

access the webpages again and change date of last access

line 258: "each 500 PAH mass spectra from ambient air particles"

500 PAH mass spectra from ambient air particles each

Table 2

Some lines are not separated by an empty line, in some lines the m/z are not aligned with the species. This has not been changed since the former version

Caption of Supplementary figure S1: "each n = 1200"

n=1200 each

Figure S5: The scale of the Meters AGL can be zoomed by about a factor of 2

Supplement line 34: "as discussed before"

as discussed in section 3.3